# Identification of the Lineage Markers and Inhibition of *DAB2* in In Vitro Fertilized Porcine Embryos

**DOI:** 10.3390/ijms21197275

**Published:** 2020-10-01

**Authors:** Jong-Nam Oh, Mingyun Lee, Gyung Cheol Choe, Dong-Kyung Lee, Kwang-Hwan Choi, Seung-Hun Kim, Jinsol Jeong, Chang-Kyu Lee

**Affiliations:** 1Department of Agricultural Biotechnology, Animal Biotechnology Major and Research Institute for Agriculture and Life Sciences, Seoul National University, Seoul 08826, Korea; ojn0505@snu.ac.kr (J.-N.O.); mg1011811@snu.ac.kr (M.L.); epicurian@snu.ac.kr (G.C.C.); culi1039@snu.ac.kr (D.-K.L.); ckh1122@snu.ac.kr (K.-H.C.); forevers122@snu.ac.kr (S.-H.K.); wlsthf0216@snu.ac.kr (J.J.); 2Institute of Green Bio Science and Technology, Seoul National University, Pyeongchang 25354, Korea

**Keywords:** pig, embryo, lineage markers, DAB2

## Abstract

Specification of embryonic lineages is an important question in the field of early development. Numerous studies analyzed the expression patterns of the candidate transcripts and proteins in humans and mice and clearly determined the markers of each lineage. To overcome the limitations of human and mouse embryos, the expression of the marker transcripts in each cell has been investigated using in vivo embryos in pigs. In vitro produced embryos are more accessible, can be rapidly processed with low cost. Therefore, we analyzed the characteristics of lineage markers and the effects of the *DAB2* gene (trophectoderm marker) in in vitro fertilized porcine embryos. We investigated the expression levels of the marker genes during embryonic stages and distribution of the marker proteins was assayed in day 7 blastocysts. Then, the shRNA vectors were injected into the fertilized embryos and the differences in the marker transcripts were analyzed. Marker transcripts showed diverse patterns of expression, and each embryonic lineage could be identified with localization of marker proteins. In *DAB2*-shRNA vectors injected embryos, *HNF4A* and *PDGFRA* were upregulated. *DAB2* protein level was lower in shRNA-injected embryos without significant differences. Our results will contribute to understanding of the mechanisms of embryonic lineage specification in pigs.

## 1. Introduction

Cells are continuously divided into diverse cell types starting from a single fertilized egg to the whole organism [1]. Early stage blastomeres have totipotency and select their fate between inner cell mass (ICM) and trophectoderm (TE) [2]. Cells in ICM separate into two lineages, epiblast (EPI) and primitive endoderm (Pri-Endo) [3]. Understanding this lineage specification provides clues to modulate the cell’s fate. Various embryonic factors help the cells to decide their lineages in the vertebrates [4]. Eggs of frog and zebrafish already have molecular gradation at fertilization, and the gradation determined the position of the axis of the embryo [5]. In mammals, the trigger of segregation does not exist at the time of fertilization. In the case of mice, heterogeneity is observed starting from the four-cell stage; this asymmetry induces cell fate decisions at the early stages [6]. Models of the blastocyst (BL) formation have been suggested and molecules and signals that regulate the lineage specialization have been identified [7,8].

Advanced techniques, e.g., RNA-seq, single cell-based analysis and genome-wide methylation profiling have been used to study embryo segregation. The dynamics of transcriptional factors were investigated by quantitative PCR (qPCR) in each embryonic cell [9]. Advanced imaging tools were used to determine detailed morphology of the BL visualize and distinguish each cell in an embryo in three dimensions (3D) [10]. Transcriptome profiles of embryonic cells have been examined by RNA-seq analysis in many mammalian species. Established lineage-specific genes were verified, and the balance of transcripts was suggested as a driving force of segregation in mice [11,12]. Three germ layers are distinguished clearly in the transcriptome and the marker genes were categorized based on the levels of transcripts in human embryos [13]. Some markers showed similar patterns to the mouse marker while other markers have completely different tendencies [14]. Single cell-based epigenetic studies monitor the changes in DNA-methylation in the early embryos. DNA-methylation is dynamic within the developmental stages, and the overall pattern is different between mouse and human [15,16]. Pigs have considerable physiological similarity to humans; thus, pigs can be used as a model to study early development [17,18].

In addition to mouse and human models, single cell-based studies have been conducted in other species. The transition of expression of the marker genes was observed by transcriptome analysis in the bovine embryo [19]. In porcine in vivo embryo, lineage-specific characteristics were identified by single-cell analysis. Using single cell qPCR, stage- and lineage-specific genes were identified, and ICM/TE cells were sorted according to the patterns of the marker gene expression [20]. Cells were categorized into lineage groups based on the single cell RNA-seq, and it has been demonstrated that certain conventional lineage markers (e.g., *CDX2* and *TEAD4*) do not correspond to markers detected in porcine embryos [21,22]. A number of studies have been investigating in vivo embryos; however, detailed information about in vitro fertilized (IVF) porcine embryos remains unavailable. Embryos can be produced from oocytes and sperm in the laboratory; thus, IVF has a number of advantages in accessibility including time and costs. Thus, to boost the lineage studies in pigs, characterization of the lineage markers is needed to be conducted in the IVF embryos.

Despite a number of similarities, early embryonic development, including TE segregation, shows considerable variability between species [23]. In mouse embryos, the YAP/TEAD4 signaling leads to cell segregation into TE through the *CDX2* expression [24]. As a result of cell polarization, *CDX2* is responsible for the allocation of TE in the early embryo [25,26]. *CDX2* is also essential in the polarization of porcine embryos and *OCT4* inhibition [27]. In cattle, *CDX2* is required for TE maintenance but does not repress *OCT4* [28]. However, in pigs, single cell analysis based on qPCR and RNA-seq demonstrated that the expression levels of *CDX2* had no differences between the ICM and TE cells, whereas the *DAB2* gene was abundantly expressed in the TE cells [20,21]. The role of the *DAB2* gene has to be investigated to understand the lineage specialization of the porcine embryo.

In this study, we characterized in vitro-fertilized porcine embryos and investigated *DAB2* gene inhibition. Initially, we investigated the stage-specific expression patterns of the marker genes and relative distribution of the marker proteins in the BLs. The expression levels of the marker genes for TE, ICM, and Pri-Endo were measured in various embryonic stages (4-cell, morula, early BL, and late BL). Then, BLs on day 7 were immunostained using a two-step double staining method to verify the relative distribution of the marker proteins. Then, we injected the shRNA vectors into the fertilized eggs. After culture, the levels of the transcript and protein expression in the embryos with or without shRNA were compared by qPCR and immunostaining. The results will help to understand the lineage-related profiles of IVF embryos and the role of *DAB2* in the development of preimplantation embryos.

## 2. Results

### 2.1. Marker Gene Expression of IVF Embryos during Development and Distribution of Marker Proteins in Day 7 IVF Embryos

Recent studies of porcine embryo had verified a number of common markers and discovered pig-specific lineage markers [20,21,22]. We have identified the trends of the marker gene expression in the porcine embryos (Figure 1A). Embryos in the 4-cell, morula, early BL, and late BL stages were used on days 2, 4, 5, and 7 of the culture, respectively. TE markers, including *DAB2*, *GATA3*, and *CDX2*, showed stage-specific differences. *DAB2* expression level was the highest in the early BL stage and was significantly decreased in the late BL stage compared with that in the morula and early BL stages. *GATA3* was increased from the morula to early BL stages. *CDX2* level was also increased and remained high until the late BL stage.

We examined the relative distribution of marker proteins in day 7 BLs. *SOX2*, *NANOG*, and *OCT4* were expressed at the high levels in the nuclei; the nuclei were small and dense (Figure 1B). *DAB2* was located in the cytoplasm of the cells; *DAB2* expression was relatively uniform in the whole BL (Figure 1C). Three ICM markers had different distribution (Figure 1D). *SOX2*- and *SOX17*-positive cells were exclusive to each other, and *SOX17*-positive cells were a part of the *OCT4*-expressing group of cells.

### 2.2. Cloning of Vectors and Verification of the shRNA Expression Vectors

To produce the shRNA expressing vector, candidate shRNA sequences containing sense-loop-antisense-5U were inserted on the U6 promoter site of pENTR.hU6hH1 (Figure 2A). We selected the clones with exact sequences. The plasmid vector that expresses the *DAB2* coding region was also constructed (Figure 2B).

For verification of the shRNA-expressing vectors, each candidate vector was transfected into the porcine fetal fibroblasts (pFF) together with the *DAB2* gene expression vector and the cells were cultured for 3 days (1 day with and 2 days without DNA plus lipofectamine). Complementary DNA was synthesized from RNA extracted from cultured pFF. *DAB2* mRNA level was low in candidates 1 and 2 but did not show significant changes in candidate 3. Thus, a mixture of candidates 1 and 2 was subsequently used for embryo microinjection (Figure 2C).

To verify the effect of shRNAs in embryo, we injected vector mixture into fertilized embryos. The level of *DAB2* transcript was significantly low in 4-cells embryos with shRNAs and no significant difference was observed in following stages (Figure 2D). Morula with shRNAs showed lower level of *DAB2*, but there was a lack of significant difference. In case of protein, *DAB2* level was low in 4-cells and morula stage embryos with shRNAs (Figure 2E). Representative images used to measure protein level of *DAB2* were shown in Figure 2F.

### 2.3. Marker mRNA Expression and Protein Distribution in shRNA-injected Embryos

Verified vectors expressing anti-*DAB2* shRNAs were microinjected into the IVF embryos. Empty vector with no shRNA sequence insertion was used as a control of microinjection. There were no significant differences in the cleavage rate on day 2 and no differences in the BL formation rate on day 7 (Figure 3A).

We measured the mRNA expression levels of the marker genes in day 7 BLs with or without shRNA microinjection (Figure 3B). There were no significant differences between the BLs with and without the injection of the control vector in most of the genes. With regard to the TE markers, the level of *CDX2* was lower in shRNA-injected embryos than that in embryos without the injection; however, the embryos injected with the control vector had a low level of *CDX2*. In the case of other TE markers, *DAB2* and *GATA3* had no differences between the treatments. The expression levels of the ICM markers, *SOX2*, *NANOG*, and *HNF4A*, was the highest in shRNA-injected embryos. However, there was a significant difference in *HNF4A*. The level of a Pri-Endo marker, *PDGFRA*, was higher in the shRNA-injected samples than that in the samples injected with the control vector; however, there was no difference in the level of *PDGFα* expression

Expression levels of the marker genes were also quantified in the injected BLs on day 5 (Figure 3C) when the expression of *DAB2* is the highest during embryonic development (Figure 1A). The ICM- and TE-specific genes did not have differences in the transcript levels. *PDGFRA*, a Pri-Endo marker, was increased in the samples injected with shRNAs. The levels of *AXIN2* and *TCF3* transcripts were measured in day 7 BLs to verify the downstream *DAB2* targets; however, no significant differences were observed (Figure 3D).

## 3. Discussion

The decision of cell fate with regard to cellular lineages is an autonomous process in the early embryo. The division of the cells based on cell fate is taking place inside of the embryo and requires specific triggers. These triggers and their mediators may be related to polarity within embryonic cells. Differences between the lineages have been investigated to identify a clear trigger of segregation [29]. Mammalian embryos have interspecies variability in lineage segregation [30]. The pig is an ideal animal model relevant to human application and information on lineage specification in pigs is important [31]. Moreover, in vivo and in vitro produced embryos have differences in a number of aspects [32,33]. Thus, the lineage segregation study of porcine IVF embryos is required. Therefore, we analyzed the stage-specific expression patterns of the marker genes and relative distribution of the marker proteins.

Triggers involved in the first segregation of ICM and TE should be present at least at the morula stage. Our results indicate that TE markers, including *DAB2*, *GATA3*, and *CDX2*, were not expressed in morula at the high levels. The expression levels of these three markers were increased during early BL (day 5). Because early BL already underwent ICM and TE, these genes may support maintenance or functional specification of the TE lineage. The transcription level of *DAB2* was significantly decreased during the transition from early to late BL; however, the other two genes were continuously expressed until the late BL stage. Apparently, *DAB2* has a role in an earlier stage compared to the roles of *GATA3* and *CDX2*. *SOX2* and *NANOG*, the ICM markers, had the highest levels of transcription during the 4-cell stage. This result means that their expression may start with embryonic genome activation. The expression of *SOX2* was sustained up until the morula stage and significantly decreased after the early BL stage. In the case of *NANOG*, the level of the transcript was dramatically decreased starting from the morula stage. However, *HNF4A* had no significant difference within the embryonic stages. In brief, *SOX2* and *NANOG* have stage-specific expression and decreasing trends during the embryonic development; the *NANOG* transcript was abundant in the earlier phases compared with the levels of *SOX2*. These two genes may be involved in the specialization of the ICM lineages. In humans and mice, *Sox2* and Nanog contribute to the maintenance characteristics of the cells derived from ICM of the embryos [34]. In the case of two Pri-Endo markers, the expression level of *PDGFα* showed a decreasing trend during embryonic development; however, there were no significant differences between stages. The level of the transcript of *PDGFRA* was significantly decreased during the late BL stage. In the mouse studies, PDGF signaling is essential for maintenance and expansion of the Pri-Endo lineage [35,36]. *PDGFRA* is one of the signal transducers that may be involved in the early specialization of the embryonic lineages in pigs. To define relative distribution of the marker proteins, we used five combinations of the markers. *DAB2* protein was localized in the cytoplasm and the transcriptional regulatory factors, *SOX2*, *SOX17*, and *OCT4*, were located in the nuclei of the cells. Unlike the TE marker DAB2, the ICM markers (*SOX2*, *NANOG*, and *OCT4*) had restricted localization in the corners of the BLs. Embryonic cells with high level of *OCT4* were located in the areas with high density nuclei. Despite high cell density, *DAB2* protein was not condensed in the ICM region. Therefore, *DAB2* level may be limited in the TE cells. We also analyzed relative localization of the ICM markers. Distribution of *SOX17* was scattered but restricted within the *OCT4*-positive cells. *SOX17*-positive cells were distinguished from the *SOX2*-positive cells. *OCT4*, *SOX2*, and *NANOG* play a role in the maintenance of the undifferentiated status in the human and mouse embryonic stem cells [37,38]. Our results suggest that *OCT4*, *SOX2*, and *SOX17* are the protein markers of ICM, EPI, and Pri-Endo in pig embryos, and this interpretation is consistent with the data obtained in the human and mouse embryos.

After validation of the knockdown ability in cultured cells, the shRNA vectors were injected into the IVF embryos at the day of fertilization. Both levels of transcript and protein were decreased with anti-*DAB2* shRNAs in 4-cell embryos, *DAB2* protein level of *DAB2* was low in shRNA injected morula. The developmental competency of the embryos was not influenced by the injection of the vectors. Injection of the shRNA did not cause reduction in the levels of the TE marker transcripts (*CDX2* and *GATA3*) including *DAB2* levels in the BLs. The transcriptional levels of *HNF4A* (an ICM marker) and *PDGFRA* (a Pri-Endo marker) were significantly increased in the shRNA-injected BLs. These markers may be more sensitive to *DAB2* than the other marker genes. Another possibility explaining this result is that the low level of *DAB2* in the earlier stage prior to day 5 leads to overexpression of *HNF4A* and *PDGFRA* on days 5 and 7. The other ICM and Pri-Endo marker genes showed an increasing trend; however, significant differences were not detected. The markers of ICM and Pri-Endo are the counterparts of TE and were upregulated without significant reduction in the levels of the *DAB2* gene. *DAB2* expression level was high during the early BL stage; hence, the lineage markers were quantified on day 5 of embryo culture. *PDGFRA* was the only tested marker whose expression was significantly higher in the shRNA-injected BL compared with that in the control. *HNF4A* is a well-known marker of definitive endoderm [39]. However, additional studies on the role of *HNF4A* in the early embryo are required to explain our results. PDGF signaling is important for the establishment of Pri-Endo [35,36]. To confirm whether embryonic *DAB2* utilizes the WNT/β-catenin singling pathway as a downstream signaling mechanism, expression levels of *AXIN2* and *TCF3*, target genes of the WNT/β-catenin singling, were measured [40]. The levels of both genes were not influenced by shRNAs. Thus, *DAB2* action may involve another pathway of the early embryo. In mouse studies, the *dab2* gene plays diverse roles in embryonic development [41]. RNA-seq of the *DAB2*-controlled embryos is required to confirm the exact embryonic pathway of *DAB2*. The level of the *DAB2* protein was reduced by shRNAs; however, the differences were not significant. This result is consistent with the data of the mRNA levels in the BL. The vectors expressing shRNA were inefficient in suppressing the expression of *DAB2*. Transcription of the plasmid vectors is getting progressively lower concomitant to the progress of the embryonic stages [42]; hence, the downregulation effect of *DAB2* may be weak in the BLs on days 5 and 7. Knockout of *dab2* is embryonically lethal in mice; thus, this gene certainly has an important role in embryonic development [43]. In single cell-based RNA studies, the quantity of *DAB2* was relatively high in the porcine TE cells [20,21]. Thus, we hypothesized that porcine *DAB2* plays an important role in embryonic development and inhibition of *DAB2* can be critical for lineage specialization. Only with the reduction of *DAB2* in the early embryonic stages, some marker genes for EPI and Pri-Endo were increased in the blastocyst stages.

In conclusion, we confirmed stage-specific expression patterns of the marker transcripts and defined distribution of the marker proteins in pig embryos. Our protocol for immunocytochemistry can be especially helpful in the analysis of the responses of embryonic lineages under various experimental conditions. Additionally, we examined the effect of anti-*DAB2* shRNAs on the lineage marker genes. *HNF4A* and *PDGFRA*, which are the ICM- and Pri-Endo-specific genes, respectively, were upregulated by shRNA-*DAB2*. We hope that our results are useful for investigation of the specialization of the embryonic lineages in pigs and may to fill the gap between the human and mouse studies.

## 4. Materials and Methods

The care and experimental use of pigs and mice were approved by the Institute of Laboratory Animal Resources, Seoul National University (SNU-140328-2). Unless otherwise stated, all chemicals were obtained from Sigma-Aldrich Corp. (St. Louis, MO, USA).

### 4.1. In Vitro Production of Fertilized Embryos

The ovaries of the prepubertal gilts were obtained from a local slaughterhouse (Anyang-si, Gyeonggi-do, Korea) and transferred to the laboratory in warm saline. Cumulus-oocyte complexes (COCs) were collected by aspirating 3- to 7-mm follicles of the prepubertal gilts with a 10-mL syringe and an 18-gauge needle. COCs with compact multiple layers (class A_1_ and A_2_ [44]) of the cumulus cells and fine cytoplasm were collected from the aspirated porcine and allowed to maturate for 44 h in the tissue culture medium 199 (Gibco, Grand Island, NY, USA) supplemented with 10% follicular fluid, L-cysteine (0.1 mg/mL), sodium pyruvate (44 ng/mL), epidermal growth factor (10 ng/mL), insulin (1 mg/mL), and kanamycin (75 μg/mL) at 39 °C. The COCs were matured with 10 IU/mL of gonadotropin hormone, pregnant mare serum gonadotropin (Lee Biosolutions, Maryland Heights, MO, USA), and human chorionic gonadotropin for the first 22 h. After the maturation, cumulus cells were removed from the oocytes with hyaluronidase. Class Ⅱ oocytes were selected for further experiments [45]. Fresh semen with high viability and motility was delivered every week from Darby Genetics Inc. (Anseong, Gyeonggi-do, Korea). Sperm was washed twice with 0.1% bovine serum albumin (BSA) supplemented with Dulbecco’s phosphate buffered saline (DPBS) at 1400 rpm for 3 min. Washed sperm was coincubated with the matured oocytes in droplets of a modified Tris-buffered medium (mTBM) and covered with mineral oil for 6 h (Abeydeera and Day, 1997). Each drop contained 50 μL total medium, 20–25 oocytes, and 2 × 10^5^/mL of sperm. mTBM was composed of 113.1 mM sodium chloride, 3 mM potassium chloride, 7.5 mM calcium chloride, 20 mM Trizma^®^ base, 11 mM glucose, 5 mM pyruvate, 1 mM caffeine, and 0.8% BSA. After this process, the eggs were incubated in 5% CO_2_ and 5% O_2_ at 39 °C in 20 μL of the porcine zygote medium 3 [46]. The cleavage rate was measured on day 2 after the insemination.

### 4.2. RNA Extraction and Quantitative PCR

Embryonic RNA was extracted by an Arcturus^®^ PicoPure^®^ RNA isolation kit (Applied Biosystems, Foster City, California, USA) according to the standard protocol. cDNA was synthesized from the total RNA of a single embryo with a high-capacity RNA-to-cDNA kit (Applied Biosystems, USA) according to the standard protocol. The levels of the transcripts were normalized to the GAPDH expression level. The list of primers is shown in Table A1.

### 4.3. Immunocytochemistry of Embryos

Embryos were washed twice with DPBS supplemented with 0.1% BSA and fixed with 4% paraformaldehyde in DPBS at room temperature (RT) for 15 min. Fixed embryos were permeabilized using 0.2% Tween-20 and 0.2% Triton X-100 in DPBS at RT for 15 min and then blocked with 10% donkey serum in DPBS at RT for 1 h. Samples were stained with anti-*SOX2*, *DAB2*, *NANOG*, *OCT4*, and *SOX17* in DPBS containing 10% donkey serum at 4 °C overnight. After washing three times in the washing solution (DPBS with 0.2% Tween-20 and 1% BSA for 10 min), the embryos were incubated with donkey anti-rabbit Alexa594 or goat anti-rabbit Alexa488 (Invitrogen, Carlsbad, California, USA) in DPBS with 10% donkey serum at RT for 1 h. For double staining, samples were stained again with the corresponding primary and secondary antibodies. Steps were the same but primary antibody treatment was conducted at RT for 2 h. All samples were washed three times with the washing solution after the secondary antibody treatment. Immunostained embryos were mounted on a slide glass with Prolong gold with DAPI (Invitrogen) and cured for at least 24 h. The list of antibodies is shown in Table A2. The imaging tool of micromanipulator was used to take the fluorescence and bright-field images. We used the ImageJ program for image processing and surface plots (monochrome images were used to produce the surface plot images).

### 4.4. Production of shRNA-Expressing Vectors and *DAB2* Gene Expression Vector

The shRNA targeting sequences within the *DAB2* coding region (Gene ID number; 100519746) were identified using an online tool (https://www.thermofisher.com). The shRNA sequences were aligned to the whole transcriptome of the pig through the BLAST program (https://blast.ncbi.nlm.nih.gov/Blast.cgi). Three sequences with an off-target low level were used to produce the DNA constructs containing the targeting sequence and a nine-nucleotide loop. Pairs of the synthesized oligonucleotides were dimerized by slow cooling from 95 to 4 °C and inserted into the U6 cloning site of the pENTR.hU6hH1 plasmid vector. The Dab2 coding region sequence was synthesized and inserted into the multiple cloning site of the pcDNA3-EGFP vector. All vectors were verified by nucleotide sequencing.

### 4.5. Culture of Porcine Fetal Fibroblasts and Plasmid Transfection

Basic cell culture and lipofection were carried out as described in our previous report [47]. Briefly, the vectors containing shRNA and expressing *DAB2* gene at a 1:1 ratio were introduced into pFF using Lipofectamine 3000 reagent (Thermo Fisher Scientific, Waltham, MA, USA). We replaced the culture medium with fresh Dulbecco’s modified Eagle’s medium 24 h after from lipofection and the cells were subsequently cultured for 2 days. The total RNA of the transfected pFFs was isolated with TRIzol reagent (Invitrogen, USA) according to the manufacturer’s instructions. cDNA was synthesized with the same kit and protocol as described above. The levels of the transcripts were normalized versus the *ACTB* expression levels.

### 4.6. Microinjection of Plasmid Vectors into IVF Embryos

Microinjection procedure was conducted with a micromanipulator (Eclipse TE2000, Nikon, Tokyo, Japan) with the holding and injection pipettes. We used Femtotip Ⅱ (Eppendorf, Hamburg, Germany) as an injection pipette. The concentration of the shRNA vectors (1:1 ratio of sh-*DAB2*-1 and sh-*DAB2*-2) was 300 ng/µL. To increase the efficiency of plasmid transfer, Lipofectamine stem transfection reagent (Invitrogen) was used according to the manufacturer’s manual. Plasmid vectors with Lipofectamine (50 ng/µL) were loaded and injected into the fertilized eggs on the day of IVF.

### 4.7. Statistical Analysis

Statistical analysis of the data was performed using GraphPad Prism software (version 5.01; San Diego, CA, USA). Significant differences in the experimental groups were determined by one-way analysis of variance followed by Tukey’s multiple comparison test. A *p*-value < 0.05 was considered significant. Data are presented as the mean ± standard error.

## Figures and Tables

**Figure 1 ijms-21-07275-f001:**
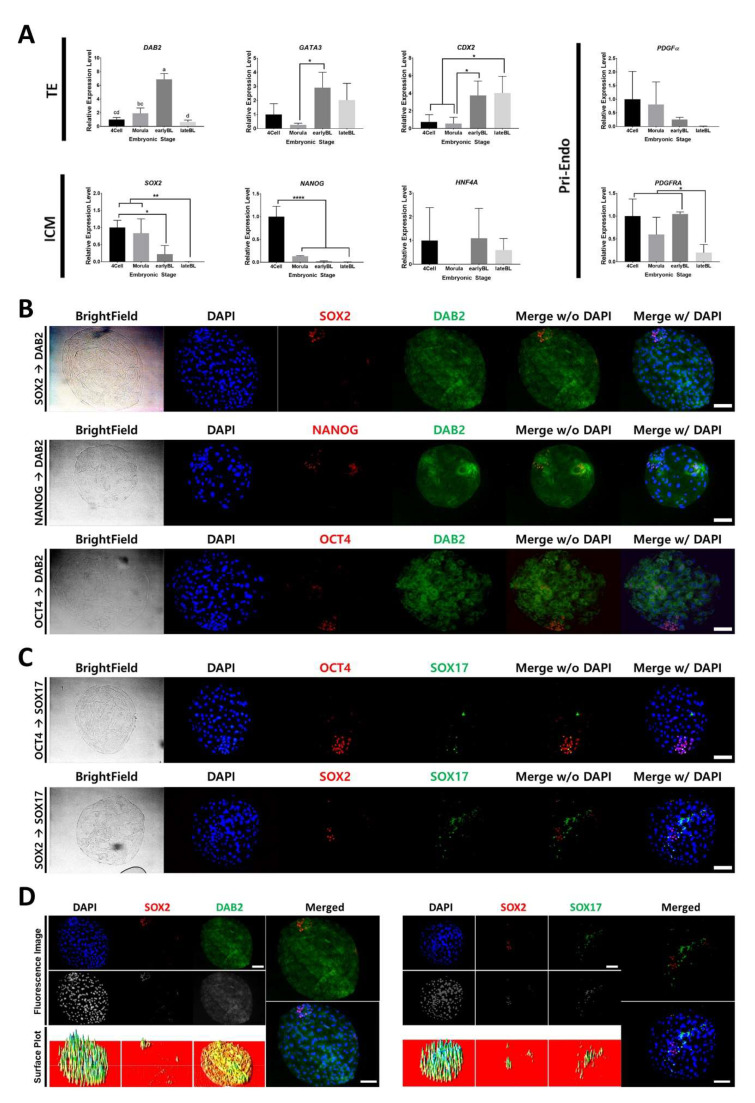
Marker mRNA levels in embryonic stages and marker protein distribution in day 7 blastocyst. (**A**) Expression levels of well-known lineage markers (Trophectoderm (TE); *DAB2*, *GATA3*, *CDX2*; Inner cell mass (ICM); *SOX2*, *NANOG*, *HNF4A*; Primitive endoderm (Pri-Endo); *PDGFα*, *PDGFRA*) were measured in 4-cell, morula, early BL and late BL stages. Different letters and * correspond to significant differences. (*: *p* < 0.05, **: *p* < 0.01, ****: *p* < 0.0001). (**B**), (**C**) and (**D**) Double immunostaining of day 7 IVF embryos. Images of bright field, DAPI and two lineage marker proteins are shown. Size marker corresponds to 100 μm. (**B**) *SOX2*, *NANOG* and *OCT4* in combination with *DAB2*. (**C**) *SOX2* and *OCT4* in combination with *SOX17*. Size markers correspond to 100 μm. (**D**) Surface plot image of Figure 1B (*SOX2* and *DAB2*) and 1C (*SOX2* and *SOX17*). Higher peak in the surface plot corresponds to the higher signal on the image. Size marker corresponds to 100 μm.

**Figure 2 ijms-21-07275-f002:**
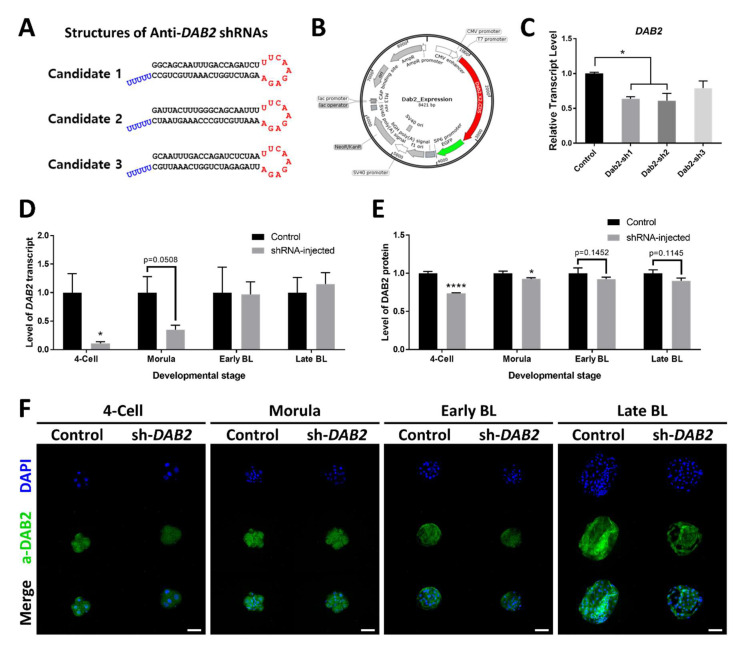
Vector information and verification of shRNA. (**A**) Expected shRNA structures of each candidate. (**B**) Plasmid map of the *DAB2* coding region expression vector. (**C**) *DAB2* transcript levels in the pFF cells transfected with shRNA vectors. (**D**) Relative level of *DAB2* transcript and (**E**) relative level of *DAB2* protein in 4-cells, morula, early blastocyst, and late blastocyst stages with or without shRNAs. (**F**) Immunocytochemistry images of *DAB2* protein in 4-cells, morula, early blastocyst, and late blastocyst stages with or without shRNAs. * corresponds to significant differences. (*: *p* < 0.05, ****: *p* < 0.0001). Size makers correspond to 100 μm.

**Figure 3 ijms-21-07275-f003:**
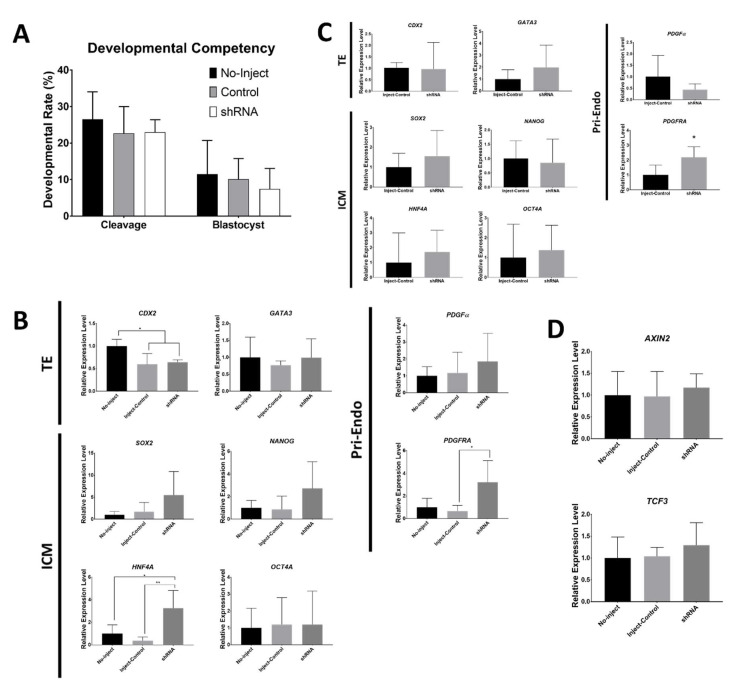
Analysis of embryos after microinjection of anti-Dab2 shRNA vector. No-inject; embryos without microinjection, Control; embryos that were injected with an empty vector without shRNA, shRNA; shRNA vector-injected embryos (1:1 mixture of candidate 1 and 2). (**A**) Development competency of embryos with or without microinjection. (**B**) Expression levels of marker mRNAs in day 7 embryos (Trophectoderm (TE); *GATA3*, *CDX2*; Inner cell mass (ICM); *SOX2*, *NANOG*, *HNF4A*; Primitive endoderm (Pri-Endo); *PDGFα*, *PDGFRA*). (**C**) Expression levels of marker mRNAs in day 5 embryos (Trophectoderm; *GATA3*, *CDX2*; Inner cell mass; *SOX2*, *NANOG*, *HNF4A*, *OCT4A*; Primitive endoderm; *PDGFα*, *PDGFRA*). (**D**) Expression levels of *DAB2* target candidates (*AXIN2* and *TCF3*) in day 7 embryos. * corresponds to significant differences. (*: *p* < 0.05, **: *p* < 0.01).

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
