# Peer review of "Identification of the Lineage Markers and Inhibition of DAB2 in In Vitro Fertilized Porcine Embryos"

_ijms, 2020, doi:10.3390/ijms21197275_

Round 1
Reviewer 1 Report
In this manuscript, the authors attempt to evaluate the specification of embryonic lineages in the pig. This is an important evaluation given the rising use of pig in research studies. Unfortunately, while the study is well conceived and designed, the rigor withwhich it is undertaken is lacking which limits the conclusions that can be made.
Major Issue:
- The authors developed an shRNA strategy to knock down DAB-2, a marker of trophectoderm that is believed to play a role in lineage specification, in Blastocysts. The authors report that the shRNA inhibition of DAB-2 was ineffective (eg expression of DAB-2 was not significantly changed), yet they report that HNF4A and PDGFRA levels were significantly altered. Without validating that the shRNA strategy was effective, the conclusions of this manuscript are not supported.
- While the authors did evaluate stage related expression of various markers of trophectoderm, ICM, and primitive endoderm, these results were largely confirmatory of the work of others, and therefore, minimally incremental in their addition to the field.
Author Response
Dear Reviewer,
Thank you for your kind comments. We prepared replies to ‘Major issue’. We hope our reply is acceptable to you.
- In our result, the level of DAB2 transcript did not show a significant difference. Protein level from ICC images was decreased in shRNA embryos, but there was no significant difference between groups. Therefore, it is hard to conclude that the up-regulation of HNF4A and PDGFRA resulted from shRNA-DAB2. However, in a case, the protein level of the target was down-regulated using RNAi without significant difference of transcript (Senechal et al., Journal of Neurochemistry, 2007). For this reason, we understood our result can be an output of a similar phenomenon. In addition, because the embryo experiment has a limitation of sample quantity, we could not conduct western blot with microinjected embryos. Nevertheless, the quantification of protein from ICC images was not successful to find significant difference. To explain the transcriptional changes of HNF4A and PDGFRA, we tried to suggest the possibilities in the section. As you pointed, down-regulation or suppression of DAB2 was not observed. Thus, we revised this part of the manuscript, following your advice.
L216-217 Removed - "Thus, DAB2 may control the upstream pathway of HNF4A and PDGFRA. "
L220 Removed - "Therefore, mild inhibition of DAB2 mRNA may induce overexpression of HNF4A and PDGFRA. "
L241-242 Removed - "DAB2 can represent the TE lineage of the preimplantation embryo in pigs. However, this gene might not be a regulator and is rather an outcome of TE segregation."
L247 Removed - "No dramatic change was observed; however, "
L249-251 Removed - "Apparently, DAB2 is not a marker of already committed TE cells but is rather an inducer of the TE lineage. Moreover, DAB2 continuously regulates the marker genes of other lineages, even after segregation."
L248 Changed - "knock down of DAB2" --> "sh-DAB2"
- As you mentioned, many researches have suggested the expression pattern of genes on embryonic stages. However, our result includes genes that recently discovered from in vivo embryos using single-cell based transcripts analysis. Information of in vitro fertilized embryo could be useful for the studies investigating specification of the embryonic lineages.
Additional changes
L44-45 Added - "in many mammalian species"
L81 Removed - "to suppress the DAB2 gene"
L81 Removed - "a period of"
L102 Change contrast and brightness of brightfield images (panel B)
L103 Changed – “during embryonic development” à “in embryonic stages”
L104-105 Abbreviation added - "Trophectoderm (TE)", "Inner cell mass (ICM)", "Primitive endoderm (Pri-Endo)"
L152 Figure revised - Relocation of panel B and C
L157-158 Abbreviation added - "Trophectoderm (TE)", "Inner cell mass (ICM)", "Primitive endoderm (Pri-Endo)"
L163 Moved to the end of the legend - "Size makers correspond to 100 μm."
L258-259 Added - "(Anyang-si, Gyeonggi-do, Republic of Korea)"
L261 Added - "(class A1 and A2)"
L268-269 Added - "Class Ⅱ oocytes were selected for further experiments"
L269-270 Added - "Fresh semen with high viability and motility was delivered every week from Darby Genetics Inc. (Anseong, Gyeonggi-do, Republic of Korea)."
L270-271 Removed - "that had been density separated or had been treated with exemestane "
L290-294 Removed – concentrations of antibodies
Table A2 - Add column of "Concentration of antibody"
Reviewer 2 Report
Manuscript ID: ijms-912543
Title: Identification of the lineage markers and inhibition of DAB2 in in vitro fertilized porcine embryos
Journal: International Journal of Molecular Sciences
Brief summary:
The study aimed to determine the characteristics of lineage markers in porcine embryos obtained after in vitro fertilization, tested the effective method of slicing the DAB2 gene in porcine fetal fibroblasts, and whether the injection of anti-Dab2 shRNA to early-stage embryos affects the expression of selected genes. First, the relative mRNA transcript abundance of lineage markers was tested. The expression level of DAB2, GATA3, CDX2, SOX2, NANOG, PDGFRA, was found to be altering in the course of early embryonic development (from 4-cell stage to late blastocyst), while the relative mRNA transcript abundance of HNF4A and PDGFA do not significantly alter in the studied stages of embryonic development. There was established an effective method of slicing the DAB2 gene in porcine fetal fibroblasts. Next, the developmental competency of porcine embryos in non-injected, control, and shRNA injected cleavage and blastocyst stage embryos were tested. Finally, the expression of DAB2 gene, the selected lineage markers, and DAB2 targeted candidates was tested in non-injected, control, and shRNA injected blastocyst at day 5 and day 7. In conclusion, in pig embryos, OCT4, SOX2, and SOX17 are the protein markers of ICM, EPI, and Pri-Endo. The injection of shRNA does not affect the developmental potential of porcine embryos. Moreover, the DAB2 gene might be involved in the regulation of HNF4A and PDGFRA expression and might be recognized as an inducer of the TE lineage and the regulator of other lineages markers expression.
However, due to not obtaining the DAB2 silencing in porcine embryos (results presented in figure 3, not altered expression of DAB2 on gene and protein level in embryos after injection of shRNA, P > 0,05), I find the last part of the conclusion overstated and not having the support in the results.
Broad comments:
The Manuscript is very interesting and in my opinion, might be found very valuable for many researchers studying the early embryonic development in pigs. The structure of the Manuscript is well organized and attractive. The information presented in the Introduction section is clear and well written what can be recognized as a strength of the Manuscript. However, it would be an advantage to highlight why it is tempting to study porcine embryos, especially that some studies were performed on humans and mice. I can see some explanation it in the second and third paragraph of the Introduction section, and in the Discussion, but still, in my opinion, it is not presented strongly enough in the Introduction. Maybe the Authors could highlight here, that pigs' embryos are a great model of human embryos and indicate why they are better than mouse embryos. The huge advantage of the manuscript is presenting the expression of the lineage markers in porcine embryos of different developmental stages, and presenting the effective method of silencing DAB2 gene expression in porcine fetal fibroblasts. These results might be especially attractive to other researchers. However, it is interesting that injection of anti-Dab2 shRNA vector to day 5 and day 7 blastocyst did not result in the decrease of DAB2 gene expression in the trophectoderm. I can see the attempt to explain this phenomenon in the discussion section, and I understand that the method of transfection used in this study was used by the Authors also previously, so they probably do not have objections for the method. Nevertheless, I would like to ask whether the Authors have been testing the effectiveness of Lipofectamine 3000 reagent in their studies. In my colleague's experience, the previous version of this product (Lipofectamine 2000) was not effective enough to introduce siRNA to some porcine cell lines. This could be tested for example by using siGLO reagent. Did the Authors consider using it (or other reagents like that) to test the effectiveness of transfection? Maybe, the lack of silencing the DAB2 gene in day 5 and day 7 blastocysts, results from the ineffective transfection in the case of this biological material, despite the promising results in porcine fetal fibroblasts. The not altered DAB2 expression on gene and protein level in control and injected blastocysts (which is presented in Fig 3, P > 0.05) in my opinion, stands for an ineffective transfection process. The effectiveness of the transfection put a bias on the results obtained within the analysis of embryos after microinjection of anti-Dab2 shRNA vector and put this section of the research really doubtful. In fact, the Authors did not achieve the DAB2 silencing in tested blastocysts. The not altered DAB2 gene expression in control and injected blastocysts, in my opinion, do not allow us to conclude about the effect of DAB2 on the expression of other genes. This is the major weakness of this work and is the reason for selecting the major revision as my recommendation. After addressing this broad comment, the Discussion section would probably need some improvements.
Apart from the above, there appeared some minor weaknesses, which I present in the Specific comments section.
Specific comments
- Title page: I would like to ask whether the Authors are sure that in the title there should be an expression: “in vitro fertilized embryos”? I guess that “in vitro fertilized oocytes” or “in vitro obtained embryos” or “porcine embryos obtained after in vitro fertilization” would be more appropriate. Also, please note that the title of the Manuscript placed in the system is different than the title placed on the first page of the Manuscript. It needs unifying.
- P2, L44: „Transcriptome profiles…” Could you specify in which species?
- P2, L81: I suggest removing “a period of”. “After the culture” expression would be clear enough in this section
- Figure 1 Panel A: In the description of the figure the Authors use full names, i.e. Trophectoderm, Inner cell mass, Primitive endoderm, and in Figure 1 the Authors use abbreviations. I believe that introducing abbreviations to Figure 1 legend next to full names would be an advantage, especially that at the end of the Manuscript the Authors show the list of abbreviations.
- Figure 1 Panel A: Figure legend introduces misunderstanding. In the first sentence of the legend, it is written: “….day 7 blastocyst”, while panel A corresponds to different stages of embryonic development. The Authors specify it in the text, but I believe that making an effort to make the figure and the figure legend could be an advantage and allow avoiding misunderstandings.
- Figure 1 Panel B: Could you increase the contrast of the brightfield – even in color view it is not visible, especially in the case of SOX2à DAB2. In other cases, I can see the blastocysts in brightfield only after high magnification and only because I know that I should find something right there.
- Figure 3 Panels B, C, D: In the description of the figure the Authors use full names, i.e. Trophectoderm, Inner cell mass, Primitive endoderm, and in Figure 3 the Authors use abbreviations. I believe that introducing abbreviations to Figure 3 legend next to full names would be an advantage, especially that at the end of the Manuscript the Authors show the list of abbreviations.
- Figure 3: This figure is very hard to follow and really needs editing to make it clear and easy-readable. First, the organization of panels is messy – in the first look of the reader, the graphs presented in panel C and panel D look like there is only one panel. Maybe the Authors could find a better way to better separate these panels. In my opinion, panel C should stand instead of panel B (in the right upper corner – because it represents embryos of day 5), the panel B should be placed below panel A, in the middle on the left side and next to it, i.e. in the middle right, should stand panel D – because these two panels represent embryos of day 7, but there are different markers analyzed. The description of size markers in Figure 3 legend could be moved to the end of this legend.
- P10, L259: Please indicate more specific information concerning the slaughterhouse (city).
- P10, section “In vitro production of fertilized embryos”. Could you please indicate the class of the oocytes? Some advice you can find in these papers: Alverez G. et al. Immature oocyte quality and maturational competence of porcine cumulus-oocyte complexes subpopulations. Biocell 2009 33(3), 167-177, DOI: 10.32604/biocell.2009.33.167, and Hiraga K. et al. Selection of in vitro-matured porcine oocytes based on localization patterns of lipid droplets to evaluate developmental competence. J Reprod Dev. 2013;59(4):405-408. DOI:10.1262/jrd.2012-126
- P10, section “In vitro production of fertilized embryos”. In my opinion, there is missing information about whether the sperm was tested for vitality, morphology, and motility. If these tests have been performed, it would be recommended to place the description of criteria and used tests in the Manuscript. If this kind of test has been not performed, please explain why. This information is important to make the experiment reproducible.
- P10, section “RNA extraction, and quantitative PCR” could you indicate how many embryos were used for RNA extraction? (number/ total weight/ or other parameter). It would increase the reproducibility of the experimental procedures.
- P10, section “RNA extraction and quantitative PCR” and P11, section “Culture of porcine fetal fibroblasts and plasmid transfection”. Could you please explain why for normalization of transcript abundance were used different housekeeping genes? Why not using the geometric mean of the expression of these two selected housekeeping genes to normalize tested genes transcript abundance? Have you been testing the stability of the expression of selected housekeeping genes?
- P10, section “Immunocytochemistry of embryos”. Could you indicate what was the negative control for staining?
- P10, section “Immunocytochemistry of embryos”. I recommend transferring the concentration of reagents to Table A2. It would help other researchers to find the relevant experimental procedures and protocols for their experiments.
- P11, L306: There appeared some typographical inconsistencies in describing thermal conditions. Please unify with other parts of the Manuscript. This is only a small help in editing.
- Please add “pFF” to the abbreviation list. This is only a small help in editing.
Author Response
Dear Reviewer,
Thanks for your kind and detailed review. All of your comments are helpful to make the article better. We described replies for the ‘Broad comments’ below. Almost of the advices in ‘Specific comments’ were applied on the manuscript and figures and the answers are also described one by one for the other comments. We hope our reply is acceptable to you.
- In our result, the level of DAB2 transcript did not show a significant difference. Protein level from ICC images was decreased in shRNA embryos, but there was no significant difference between groups. Therefore, it is hard to conclude that the up-regulation of HNF4A and PDGFRA resulted from shRNA-DAB2. However, in a case, the protein level of the target was down-regulated using RNAi without significant difference of transcript (Senechal et al., Journal of Neurochemistry, 2007). For this reason, we understood our result can be an output of a similar phenomenon. In addition, because the embryo experiment has a limitation of sample quantity, we could not conduct western blot with microinjected embryos. Nevertheless, the quantification of protein from ICC images was not successful to find significant difference. To explain the transcriptional changes of HNF4A and PDGFRA, we tried to suggest the possibilities in the section. As you pointed, down-regulation or suppression of DAB2 was not observed. Thus, we revised this part of the manuscript, following your advice.
L216-217 Removed - "Thus, DAB2 may control the upstream pathway of HNF4A and PDGFRA. "
L220 Removed - "Therefore, mild inhibition of DAB2 mRNA may induce overexpression of HNF4A and PDGFRA. "
L241-242 Removed - "DAB2 can represent the TE lineage of the preimplantation embryo in pigs. However, this gene might not be a regulator and is rather an outcome of TE segregation."
L247 Removed - "No dramatic change was observed; however, "
L249-251 Removed - "Apparently, DAB2 is not a marker of already committed TE cells but is rather an inducer of the TE lineage. Moreover, DAB2 continuously regulates the marker genes of other lineages, even after segregation."
L248 Changed - "knock down of DAB2" --> "sh-DAB2"
- “Effectiveness of transfection into the embryo.” - There is a report that liposome helps expression of plasmid vector in microinjection of the bovine embryo (Vichera et al., Reprod Dom Anim, 2011). We are using this method, and the induction of vector expression is successful in current experiments. Therefore, in our humble opinion, the transfection protocol does not have a problem.
Amyloid precursor protein knockdown by siRNA impairs spontaneous alternation in adult mice, Yann Senechal, Peter H. Kelly, John F. Cryan, Francois Natt and Kumlesh K. Dev, Journal of Neurochemistry, 102, 1928–1940 (2007)
Efficient Transgene Expression in IVF and Parthenogenetic Bovine Embryos by Intracytoplasmic Injection of DNA–Liposome Complexes, G Vichera, L Moro and D Salamone, Reprod Dom Anim 46, 214–220 (2011)
Minor changes
L44-45 Added - "in many mammalian species"
L81 Removed - "to suppress the DAB2 gene"
L81 Removed - "a period of"
L102 Change contrast and brightness of brightfield images (panel B)
L103 Changed – “during embryonic development” à “in embryonic stages”
L104-105 Abbreviation added - "Trophectoderm (TE)", "Inner cell mass (ICM)", "Primitive endoderm (Pri-Endo)"
L152 Figure revised - Relocation of panel B and C
L157-158 Abbreviation added - "Trophectoderm (TE)", "Inner cell mass (ICM)", "Primitive endoderm (Pri-Endo)"
L163 Moved to the end of the legend - "Size makers correspond to 100 μm."
L258-259 Added - "(Anyang-si, Gyeonggi-do, Republic of Korea)"
L261 Added - "(class A1 and A2)"
L268-269 Added - "Class Ⅱ oocytes were selected for further experiments"
L269-270 Added - "Fresh semen with high viability and motility was delivered every week from Darby Genetics Inc. (Anseong, Gyeonggi-do, Republic of Korea)."
L270-271 Removed - "that had been density separated or had been treated with exemestane "
L290-294 Removed – concentrations of antibodies
Table A2 - Add column of "Concentration of antibody"
Replies to Specific comments
- Title page: I would like to ask whether the Authors are sure that in the title there should be an expression: “in vitrofertilized embryos”? I guess that “in vitro fertilized oocytes” or “in vitro obtained embryos” or “porcine embryos obtained after in vitro fertilization” would be more appropriate. Also, please note that the title of the Manuscript placed in the system is different than the title placed on the first page of the Manuscript. It needs unifying.
– “In vitro fertilized embryo” is commonly used term in our field, therefore, we the title in the manuscript does not need to be edited. However, as you pointed, we will ask to change title on the system.
- P2, L44: „Transcriptome profiles…” Could you specify in which species? - We applied it on the manuscript.
- P2, L81: I suggest removing “a period of”. “After the culture” expression would be clear enough in this section - We applied it on the manuscript.
- Figure 1 Panel A: In the description of the figure the Authors use full names, e.Trophectoderm, Inner cell mass, Primitive endoderm, and in Figure 1 the Authors use abbreviations. I believe that introducing abbreviations to Figure 1 legend next to full names would be an advantage, especially that at the end of the Manuscript the Authors show the list of abbreviations. - We applied it on the figure legend.
- Figure 1 Panel A: Figure legend introduces misunderstanding. In the first sentence of the legend, it is written: “….day 7 blastocyst”, while panel A corresponds to different stages of embryonic development. The Authors specify it in the text, but I believe that making an effort to make the figure and the figure legend could be an advantage and allow avoiding misunderstandings. - We applied it on the figure legend.
- Figure 1 Panel B: Could you increase the contrast of the brightfield – even in color view it is not visible, especially in the case of SOX2à DAB2. In other cases, I can see the blastocysts in brightfield only after high magnification and only because I know that I should find something right there. - We applied it on the manuscript.
- Figure 3 Panels B, C, D: In the description of the figure the Authors use full names, e.Trophectoderm, Inner cell mass, Primitive endoderm, and in Figure 3 the Authors use abbreviations. I believe that introducing abbreviations to Figure 3 legend next to full names would be an advantage, especially that at the end of the Manuscript the Authors show the list of abbreviations. - We applied it on the figure legend.
- Figure 3: This figure is very hard to follow and really needs editing to make it clear and easy-readable. First, the organization of panels is messy – in the first look of the reader, the graphs presented in panel C and panel D look like there is only one panel. Maybe the Authors could find a better way to better separate these panels. In my opinion, panel C should stand instead of panel B (in the right upper corner – because it represents embryos of day 5), the panel B should be placed below panel A, in the middle on the left side and next to it, e. in the middle right, should stand panel D – because these two panels represent embryos of day 7, but there are different markers analyzed. The description of size markers in Figure 3 legend could be moved to the end of this legend. – Applied. We changed the location of panel B and panel C on Figure 3.
- P10, L259: Please indicate more specific information concerning the slaughterhouse (city). - We applied it on the manuscript.
- P10, section “In vitro production of fertilized embryos”. Could you please indicate the class of the oocytes? Some advice you can find in these papers: Alverez G. et al. Immature oocyte quality and maturational competence of porcine cumulus-oocyte complexes subpopulations. Biocell2009 33(3), 167-177, DOI: 10.32604/biocell.2009.33.167, and Hiraga K. et al. Selection of in vitro-matured porcine oocytes based on localization patterns of lipid droplets to evaluate developmental competence. J Reprod Dev. 2013;59(4):405-408. DOI:10.1262/jrd.2012-126 – Thanks for your kind help, we supplemented information about condition of COCs and oocytes.
- P10, section “In vitro production of fertilized embryos”. In my opinion, there is missing information about whether the sperm was tested for vitality, morphology, and motility. If these tests have been performed, it would be recommended to place the description of criteria and used tests in the Manuscript. If this kind of test has been not performed, please explain why. This information is important to make the experiment reproducible. – Applied. We used commercial fresh semen from a company, so we added sentence about the semen. This product is tested for vitality, motility, and preservability.
- P10, section “RNA extraction, and quantitative PCR” could you indicate how many embryos were used for RNA extraction? (number/ total weight/ or other parameter). It would increase the reproducibility of the experimental procedures. – We used single embryo for each sampling procedure of RNA extraction. You may find the word “single embryo” in the middle of the section.
- P10, section “RNA extraction and quantitative PCR” and P11, section “Culture of porcine fetal fibroblasts and plasmid transfection”. Could you please explain why for normalization of transcript abundance were used different housekeeping genes? Why not using the geometric mean of the expression of these two selected housekeeping genes to normalize tested genes transcript abundance? Have you been testing the stability of the expression of selected housekeeping genes? – We usually use ACTB as a housekeeping gene in the quantification of transcript in cells. However, we found that GAPDH is more stable in embryos than ACTB. That’s why we used different housekeeping genes for normalization.
- P10, section “Immunocytochemistry of embryos”. Could you indicate what was the negative control for staining? – We treated secondary antibodies without primary antibody on embryos as a negative control.
- P10, section “Immunocytochemistry of embryos”. I recommend transferring the concentration of reagents to Table A2. It would help other researchers to find the relevant experimental procedures and protocols for their experiments. - We applied it on the manuscript.
- P11, L306: There appeared some typographical inconsistencies in describing thermal conditions. Please unify with other parts of the Manuscript. This is only a small help in editing. - We applied it on the manuscript.
Please add “pFF” to the abbreviation list. This is only a small help in editing. – pFF is already on the list of abbreviation. We highlighted the line of the “pFF” on the list.
Round 2
Reviewer 1 Report
These studies continue to be flawed by technical problems. The shRNA did not work and therefore no valid conclusions can be reached.
Instead of redoing the experiments, the authors have, unfortunately, tried to explain why speculating about these flawed results may be of interest to a reader. This is unacceptable.
Author Response
Dear Reviewer,
Thank you for your review comments. We understood your concerns, so we planned to conduct further experiments to ensure the effect of shRNA on the embryo. We will measure mRNA and protein levels after shRNA injection at the stage of 4-cell, morula, and early blastocyst. We hope the following results will make you willing to accept.
Sincerely,
Chang-Kyu Lee
Reviewer 2 Report
Dear Authors,
I am pleased to inform you that you have successfully and sufficiently managed with all my comments and I find this article highly improved. Currently, since many significant changes in the Discussion section have been made, I find the conclusions being supported by the results. Thus, in my opinion, the current form of the Manuscript does not raise any doubts and I accept it in the current form. Additionally, I would like to highlit that the changes in the Figures and table are very nice and have made them easy to follow. The added information in the Materials and Methods section is sufficient and made the research reproducible. I believe that the research presented in the Manuscript might be helpful for other researchers in the future.
Please accept my congratulations and kind regards.
Author Response
Dear Reviewer,
Thank you again for your kind comments. However, there was some concern about the outcome of shRNA. To verify the effect of shRNA on embryos, we are planning, and are currently conducting further experiments. We will measure mRNA and protein levels after shRNA injection at the stage of 4-cell, morula, and early blastocyst. We hope the following results will make you willing to accept.
Sincerely,
Chang-Kyu Lee